# The plant-specific transcription factors CBP60g and SARD1 are targeted by a *Verticillium* secretory protein VdSCP41 to modulate immunity

Jun Qin[1,2], Kailun Wang[1,2], Lifan Sun[1], Haiying Xing[1], Sheng Wang[1,2], Lin Li[3], She Chen[3], Hui-Shan Guo[1,2]*, Jie Zhang[1]*

[1]State Key Laboratory of Plant Genomics, Institute of Microbiology, Chinese Academy of Sciences, Beijing, China; [2]University of Chinese Academy of Sciences, Beijing, China; [3]National Institute of Biological Sciences, Beijing, China

**Abstract** The vascular pathogen *Verticillium dahliae* infects the roots of plants to cause *Verticillium* wilt. The molecular mechanisms underlying *V. dahliae* virulence and host resistance remain elusive. Here, we demonstrate that a secretory protein, VdSCP41, functions as an intracellular effector that promotes *V. dahliae* virulence. The *Arabidopsis* master immune regulators CBP60g and SARD1 and cotton GhCBP60b are targeted by VdSCP41. VdSCP41 binds the C-terminal portion of CBP60g to inhibit its transcription factor activity. Further analyses reveal a transcription activation domain within CBP60g that is required for VdSCP41 targeting. Mutations in both *CBP60g* and *SARD1* compromise *Arabidopsis* resistance against *V. dahliae* and partially impair VdSCP41-mediated virulence. Moreover, virus-induced silencing of *GhCBP60b* compromises cotton resistance to *V. dahliae*. This work uncovers a virulence strategy in which the *V. dahliae* secretory protein VdSCP41 directly targets plant transcription factors to inhibit immunity, and reveals CBP60g, SARD1 and GhCBP60b as crucial components governing *V. dahliae* resistance.
DOI: https://doi.org/10.7554/eLife.34902.001

*For correspondence:
guohs@im.ac.cn (H-SG);
zhangjie@im.ac.cn (JZ)

**Competing interests:** The authors declare that no competing interests exist.

## Introduction

The vascular pathogen *Verticillium dahliae* infects a broad range of plants and causes devastating diseases. *Verticillium dahliae* can survive in the form of microsclerotia in soil for over ten years (*Schnathorst, 1981*). During *V. dahliae* colonization, microsclerotia germinate and develop hyphae that, upon perception of plant roots, adhere tightly to the root surface (*Zhao et al., 2014*). A few hyphae form hyphopodia at the infection site and further differentiate into penetration pegs that penetrate into plant cells and colonize the vascular tissue (*Fradin and Thomma, 2006*; *Schnathorst, 1981*; *Vallad and Subbarao, 2008*; *Zhao et al., 2014Zhao et al., 2016*). Although a few key steps mediating these infection processes have been elucidated, the molecular mechanisms underlying *V. dahliae* virulence remain largely unknown.

Progress has been made in isolating virulence factors that are crucial for *V. dahliae* virulence. Several genes that regulate *V. dahliae* development have been characterized as contributing to virulence. *VDH1* and *VdGARP1*, which regulate microsclerotial development, are required for *V. dahliae* virulence in cotton plants (*Gao et al., 2010*; *Klimes and Dobinson, 2006*). *VdRac1* and *VdPKAC1* regulate both the development and the pathogenicity of the fungus in host plants (*Tian et al., 2015*; *Tzima et al., 2010, 2012*). VdEG1, VdEG3, VdCYP1, and VdSNF1 also function as factors that contribute to *V. dahliae* virulence as it infects cotton (*Gui et al., 2017*; *Tzima et al., 2011*; *Zhang et al., 2016*).

**eLife digest** Like animals, plants have an immune system to protect themselves from disease. When a plant detects a disease-causing microbe, proteins that serve as master regulators of its immune system activate defense-related genes. Yet some microbes can overcome these defenses and successfully infect plants. *Verticillium dahliae* is a fungus, found in soil, that infects the roots of many plants – including cotton, tomatoes and potatoes. Infection by this fungus causes the leaves to curl and discolor, and the plant to wilt.

The *V. dahliae* fungus releases, or secretes, nearly 800 proteins during an infection. Yet it remains unknown if and how any of these proteins help the fungus to infect plants. A better understanding of how *V. dahliae* impairs plant immunity to infect its hosts could give insights into ways to improve plant resistance against this fungus.

Now, Qin et al. show that a secreted protein called VdSCP41 promotes *V. dahliae* infection in both cotton and *Arabidopsis* plants. Further experiments showed that after leaving the fungus, VdSCP41 enters into the plant's own cells. Protein-protein interaction and biochemical studies then indicated VdSCP41 associates with a master immune regulator in *Arabidopsis* called CBP60g. This interaction interferes with CBP60g's ability to activate the defense-related genes.

Now that this role for VdSCP41 has been confirmed, the next step would be to see if targeting it would make plants more resistant to this fungus. One approach would be to genetically engineer plants so that they can specifically shut down, or 'silence', the fungal gene that encodes for this protein. Further experiments are required to see whether using this technique – known as host-induced gene silencing (or HIGS for short) – against VdSCP41would enhance plant resistance to *V. dahliae*. If it does prove effective, this approach may eventually reduce the need for chemical pesticides to protect crop plants.

DOI: https://doi.org/10.7554/eLife.34902.002

In addition to development-associated virulence factors, fungal pathogens deliver effectors that act as virulence factors, thereby inhibiting host defense and promoting pathogenesis. Bacterial and oomycete pathogens also deliver such effectors. The race 1 strain-specific effector Ave1 contributes to *V. dahliae* virulence on tomato plants not carrying *Ve1* (*de Jonge et al., 2012*). Necrosis and ethylene-inducing peptide 1 (Nep1)-like proteins (NLPs) (NLP1 and NLP2) secreted by *V. dahliae* strain JR2 are required for its pathogenicity in tomato and *Arabidopsis* plants (*Santhanam et al., 2013*). *VdSge1* encodes a transcriptional regulator that controls the expression of six putative effector genes. Deletion of *VdSge1* in *V. dahliae* significantly impairs its pathogenicity in tomato, suggesting an important role for secreted effectors in suppressing host immunity in *V. dahliae* (*Santhanam and Thomma, 2013*).

Nevertheless, only two secreted effectors of *V. dahliae* have been indicated to function inside the plant cell to modulate host immunity to date. VdIsc1, a *V. dahliae* effector lacking a known signal peptide, is thought to be delivered into host cells to hydrolyze a salicylic acid (SA) precursor and thereby inhibit salicylate metabolism (*Liu et al., 2014*). The small cysteine-containing protein (SCP) VdSCP7 translocates into the nucleus of plant cells to either suppress or induce defense in plants through unknown mechanisms (*Zhang et al., 2017*).

Plants are equipped with immune components to counteract *V. dahliae* virulence. Tomato *Ve1* has been identified as an effective resistance locus that recognizes Ave1 secreted by Race 1 strains (*Fradin et al., 2009*; *Schaible et al., 1951*). Genetic analyses indicated that *EDS1*, *NDR1* and *SERK3/BAK1* are required for *Ve1*-mediated resistance in both tomato and *Arabidopsis* (*Fradin et al., 2009, 2011*). Studies in cotton also showed that *GhBAK1* and *GhNDR1* are crucial components in regulating defense against *V. dahliae* (*Gao et al., 2013b*; *Gao et al., 2011*), demonstrating that both *NDR1* and *SERK3/BAK1* are required in a conserved mechanism for defense against *V. dahliae* in these plants. *GbWRKY1*, *GhSSN*, *GbERF*, *GhMLP28*, *GhMKK2* and *GbNRX1* have also been shown to be required for cotton resistance against *V. dahliae* (*Gao et al., 2011*; *Li et al., 2014a, 2016*; *Qin et al., 2004*; *Sun et al., 2014*; *Yang et al., 2015*). A comparative proteomic analysis indicated the involvement of both brassinosteroids and jasmonic acid signaling pathways in the regulation of cotton resistance to *V. dahliae* (*Gao et al., 2013a*).

Although several regulators have been identified, the mechanisms through which plants defend against *V. dahliae* remain obscure, and further investigation is required to isolate more host immune components governing *V. dahliae* resistance. Targeting key immune components is a common strategy employed by pathogenic effectors to promote pathogenicity (*Boller and He, 2009*; *Cui et al., 2009*; *Dou and Zhou, 2012*); thus, the screening of host proteins targeted by pathogenic effectors provides an efficient way to identify crucial host defense components.

The calmodulin-binding proteins (CBPs) function to bind calmodulin and thus to transduce calcium signals (*Bouché et al., 2005*). The plant-specific CALMODULIN BINDING PROTEIN 60 (CBP60) protein family contains eight family members, including CBP60a–g and SYSTEMIC ACQUIRED RESISTANCE DEFICIENT 1 (SARD1) (*Bouché et al., 2005*). CBP60g and SARD1 are two closely related members that function partially redundantly in both SA signaling and bacterial resistance (*Zhang et al., 2010*; *Wang et al., 2011*). CBP60g contains a calmodulin-binding domain (CBD) that is essential for its function in defense, whereas SARD1 does not bind calmodulin (*Wang et al., 2009Wang et al., 2011*; *Zhang et al., 2010*). Both CBP60g and SARD1 function as transcription factors that directly bind to the promoters of genes that control SA synthesis, such as *Isochorismate Synthase 1* (*ICS1*), *ENHANCED DISEASE SUSCEPTIBILITY 5* (*EDS5*), and *NON-EXPRESSOR OF PATHOGENESIS RELATED GENES1* (*NPR1*) (*Dong, 2004*; *Nawrath et al., 2002*; *Sun et al., 2015*; *Wildermuth et al., 2001*). Moreover, ChIP-seq analyses have revealed that CBP60g and SARD1 directly bind to the promoters of a number of genes, thereby regulating pathogen-associated molecular patterns (PAMPs)-triggered immunity (PTI), effector-triggered immunity (ETI) and systemic acquired resistance (SAR) (*Sun et al., 2015*), indicating their broad role in the regulation of plant immunity. In this study, we identified VdSCP41 as a virulence effector that suppresses plant immunity induced by PAMPs. VdSCP41 interacts with *Arabidopsis* CBP60g and SARD1 and modulates their transcription factor activity. The contribution of VdSCP41 to *V. dahliae* virulence is significantly reduced during the infection of the *cbp60g-1/sard1-1* double mutant. GhCBP60b, the closest protein homolog of CBP60g in cotton, is also targeted by VdSCP41 and contributes to cotton resistance against *V. dahliae*. Taken together, our findings revealed that CBP60g, SARD1 and GhCBP60b are novel components that govern *V. dahliae* resistance and that these proteins are modulated by a secretory effector VdSCP41.

## Results

### VdSCP41 contributes to *V. dahliae* virulence in *Arabidopsis* and cotton plants

Secretome analyses revealed more than 700 potential secreted proteins in *V. dahliae* (*Klosterman et al., 2011*), but to date, relatively few of these secreted proteins have been characterizes as carrying virulence function. We performed a reverse genetic screen to identify secreted proteins that are crucial for *V. dahliae* virulence. Using the newly developed USER-ATMT-DS binary vector (*Wang et al., 2016*), we constructed 56 gene deletion mutants in *V. dahliae* strain 592 (V592), each of which targets an individual potentially secreted protein. The resulting mutants were subjected to virulence assessment in host plants, including upland cotton (*Gossypium hirsutum*) and the model plant *Arabidopsis*. A mutant carrying a targeted deletion of *VdSCP41* (VdΔ*scp41*) was isolated (*Figure 1A*) and shown to display significantly reduced virulence compared with WT strain V592.

*VdSCP41* encodes a hypothetical protein in the secretome of the *V. dahliae* Ls.17 strain (VdLs.17) (*Klosterman et al., 2011*). The expression of this gene is significantly upregulated at 2 days post-inoculation (dpi) in *Arabidopsis* (*Figure 1—figure supplement 1*), suggesting a putative function of *VdSCP41* in *V. dahliae* infection. Targeted gene deletion of *VdSCP41* in the VdΔ*scp41* mutant was verified by southern blotting (*Figure 1B*). The VdΔ*scp41* mutant displayed much weaker disease symptoms than WT V592 in both upland cotton (*Figure 1C*) and *Arabidopsis* (*Figure 1D*). Disease index analyses indicated significantly reduced virulence of the VdΔ*scp41* mutant compared with that of V592 in both upland cotton and *Arabidopsis* (*Figure 1E–F*, *Figure 1—source data 1*). The reduced virulence of the VdΔ*scp41* mutant was restored upon complementation with GFP-tagged *VdSCP41* (VdΔ*scp41/VdSCP41-GFP*) (*Figure 1C–F*). Thus, VdSCP41 functions as a virulence effector that contributes to *V. dahliae* virulence on host plants.

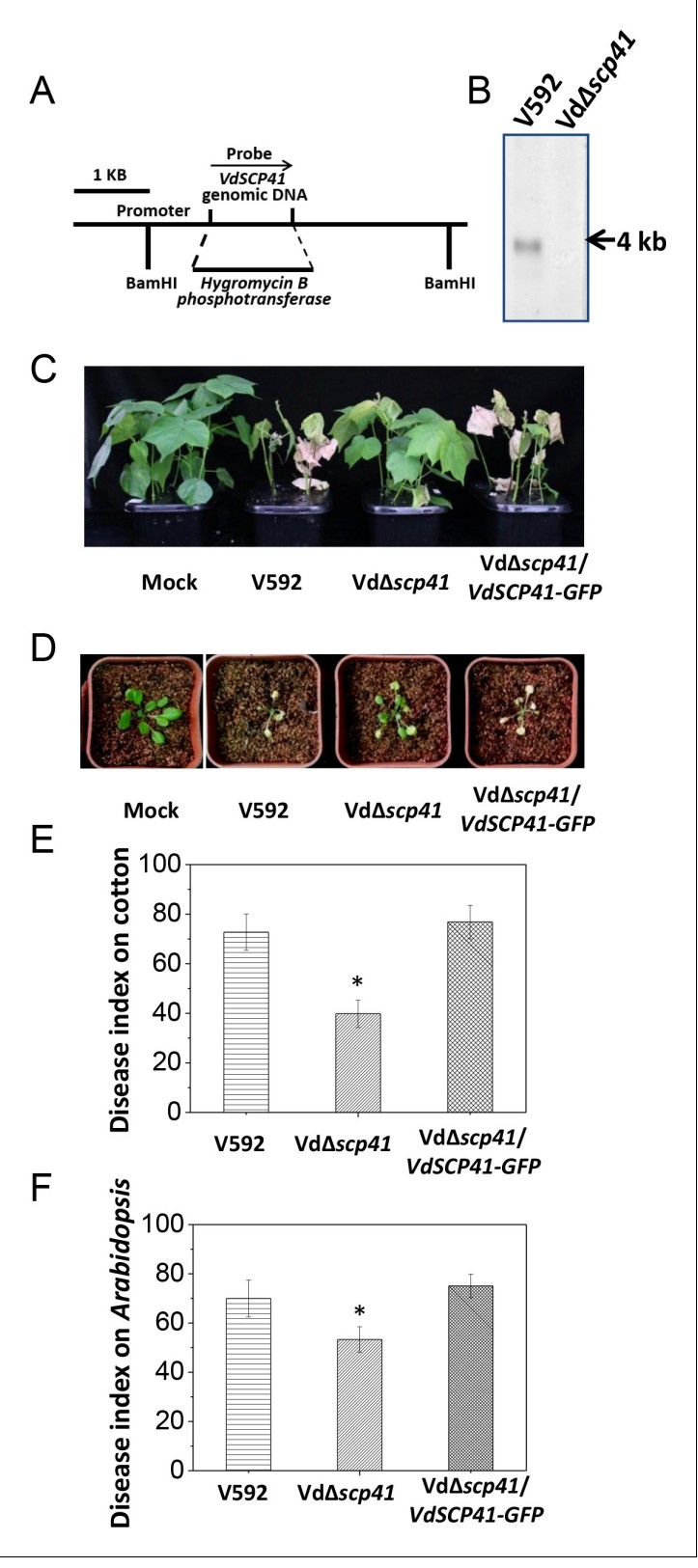

**Figure 1.** *VdSCP41* contributes to *V. dahliae* virulence in host plants. (**A**) Schematic description of the generation of the VdΔ*scp41* mutant. (**B**) Southern blot analysis of the *VdSCP41* gene deletion in the VdΔ*scp41* mutant. Genomic DNA samples isolated from V592 and VdΔ*scp41* strains were digested by *BamHI* and subjected to Southern blot analysis. (**C–D**) Disease symptoms of upland cotton (**C**) and *Arabidopsis* (**D**) plants infected with the
*Figure 1 continued on next page*

*Figure 1 continued*

wildtype (V592), VdΔ*scp41*, and VdΔ*scp41/VdSCP41-GFP* strains. (**E–F**) Disease index analyses of upland cotton (**E**) and *Arabidopsis* (**F**) infected with the indicated strains. The plants were photographed and subjected to disease index analyses 3–4 weeks post inoculation. The disease indexes were evaluated with three replicates generated from 24 plants for each inoculum. Error bars indicate the standard deviation of three biological replicates. Student's t-test was carried out to determine the significance of difference. *Indicates significant difference at a *P*-value of < 0.05. The experiments were repeated three times with similar results.

DOI: https://doi.org/10.7554/eLife.34902.003

The following source data and figure supplement are available for figure 1:

**Source data 1.** Source data for *Figure 1*.

DOI: https://doi.org/10.7554/eLife.34902.005

**Figure supplement 1.** Expression of *VdSCP41* in *V. dahliae* is induced by plant roots.

DOI: https://doi.org/10.7554/eLife.34902.004

## VdSCP41 acts as an intracellular effector

In *V. dahliae*, some signal-peptide-containing SCPs are delivered to the septin-organized hyphal neck, which develops from the base of the hyphopodia and functions as a fungus–host penetration interface for the dynamic delivery of secretory proteins (*Zhou et al., 2017*). VdSCP41 contains an N-terminal signal peptide predicted by SignalP (http://www.cbs.dtu.dk/services/SignalP/) (*Figure 2—figure supplement 1A*). Therefore, we examined the localization of VdSCP41 in *V. dahliae*. GFP-tagged VdSCP41 (VdSCP41-GFP) was found to localize to the base of the hyphopodium and showed ring signals surrounding the hyphal neck when the VdΔ*scp41* mutant strain complemented with VdSCP41-GFP (VdΔ*scp41/VdSCP41-GFP*) was cultured on a cellophane membrane for hyphopodium induction (*Figure 2A*). By contrast, VdSCP41 lacking signal peptide (ΔspVdSCP41-GFP) showed diffused signal at the base of the hyphopodium without clear ring signals surrounding the hyphal neck. The results demonstrate that VdSCP41-GFP was delivered to the penetration interface for secretion.

Nuclear localization signal (NLS) sequence prediction (http://nls-mapper.iab.keio.ac.jp/cgi-bin/NLS_Mapper_form.cgi) identified a potential NLS in VdSCP41 (*Figure 2—figure supplement 1A*). We therefore took advantage of this NLS to examine the putative translocation of *V. dahliae*-delivered VdSCP41 into the nucleus of plant cells. The VdΔ*scp41/VdSCP41-GFP* and GFP-expressing V592 (V592-GFP) strains were separately inoculated onto onion epidermal cells. Although GFP fluorescence from the V592-GFP strain was observed in conidial spores, VdSCP41-GFP secreted by *V. dahliae* was capable of translocating into plant cells, and of localizing to the nucleus in addition to the pericellular space of onion epidermal cells (*Figure 2—figure supplement 1B*). The potential NLS sequence of SCP41 was mutated (SCP41$_{-nls}$-GFP) and then complemented into VdΔ*scp41* to construct the VdΔ*scp41/VdSCP41$_{-nls}$-GFP* strain. In contrast to VdSCP41-GFP, VdSCP41$_{-nls}$-GFP failed to translocate into the nucleus of onion epidermal cells (*Figure 2—figure supplement 1B*). These results suggested that VdSCP41 delivered by *V. dahliae* translocates into the nucleus of the plant cell, which requires the NLS predicted within its sequence.

We next transiently expressed mCherry-tagged VdSCP41 in plants to further verify its nuclear localization in plant cells. As the signal peptide located at the N terminus may guide secreted proteins into the plant extracellular space in some cases, we fused mCherry to VdSCP41 both with and without (Δ*sp*VdSCP41) the signal peptide in order to analyze subcellular localization. VdSCP41-mCherry and Δ*sp*VdSCP41-mCherry were individually transiently expressed in either *Arabidopsis* protoplasts or in *Nicotiana benthamiana* (*N. b.*) leaves. mCherry fluorescence imaging revealed that both VdSCP41 and Δ*sp*VdSCP41 localized to the nucleus in *Arabidopsis* cells (*Figure 2B*). The protein expression level of VdSCP41-mCherry and Δ*sp*VdSCP41-mCherry was detected by immunoblot (*Figure 2—figure supplement 1C*). Similar nuclear localization was observed for both VdSCP41-mCherry and Δ*sp*VdSCP41-mCherry in *N. b.* cells (*Figure 2—figure supplement 1D*). These results are consistent with the nuclear localization of the *V. dahliae*-delivered VdSCP41 in onion epidermal cells.

## *VdSCP41* expression in *Arabidopsis* inhibits plant immunity

It is believed that the initial function of a fungal effector protein is to suppress PTI. To investigate whether VdSCP41 is capable of inhibiting plant immunity when it is directly expressed in plants,

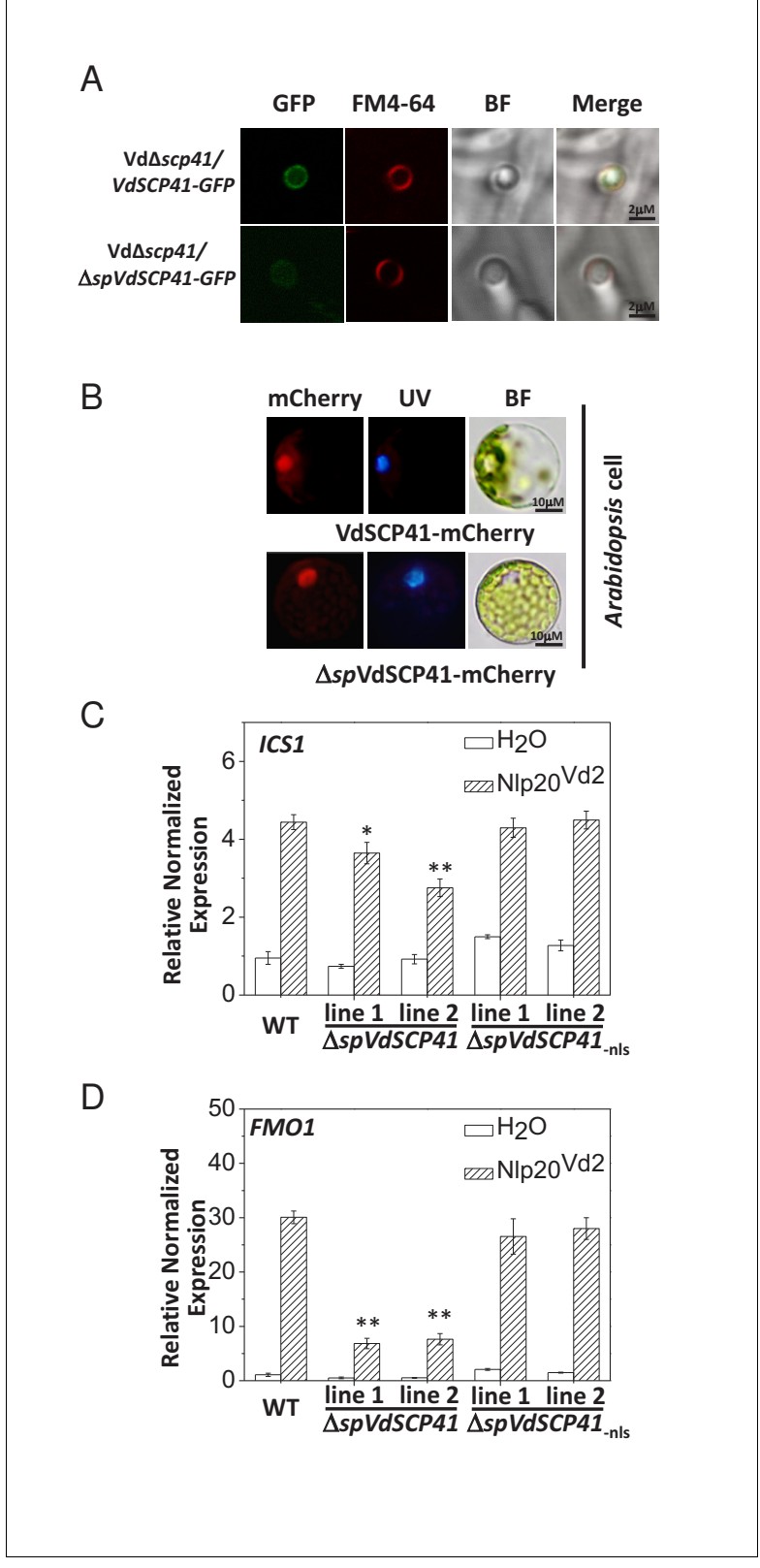

**Figure 2.** VdSCP41 functions to inhibit immunity in plants. (**A**) VdSCP41 localizes to the base of the hyphopodia and forms ring signals surrounding the hyphal neck. The VdΔ*scp41/VdSCP41-GFP* strain and VdΔ*scp41/Δ spVdSCP41-GFP* was cultured on a cellophane membrane for 5 days to induce the formation of hyphopodia. Localization of VdSCP41-GFP was visualized using a Leica SP8 microscope. (**B**) Transiently expressed VdSCP41

*Figure 2 continued on next page*

*Figure 2 continued*
preferentially localizes to the nucleus of *Arabidopsis* cells. *Arabidopsis* protoplasts were transfected with either a VdSCP41-mCherry or a ΔspVdSCP41-mCherry plasmid as indicated. mCherry fluorescence was visualized 16 hr post transfection. DAPI staining of the nucleus was visualized under UV light. (**C–D**) Expression of ΔspVdSCP41 in *Arabidopsis* inhibits nlp20[Vd2]-induced *ICS1* (**C**) and *FMO1* (**D**) expression. Wildtype (WT) and transgenic plants expressing ΔspVdSCP41 or ΔspVdSCP41[-nls] were treated with $H_2O$ or nlp20[Vd2] as indicated. RNA was extracted for real-time PCR analyses. The experiments were repeated three times with similar results. Error bars indicate standard deviations. Student's t-test was carried out to determine the significance of difference. * Indicates significant difference at a *P*-value of < 0.05, whereas ** indicates significant difference at a *P*-value of < 0.01.
DOI: https://doi.org/10.7554/eLife.34902.006
The following source data and figure supplements are available for figure 2:

**Source data 1.** Source data for *Figure 2*.
DOI: https://doi.org/10.7554/eLife.34902.009
**Figure supplement 1.** VdSCP41 delivered by *V. dahliae* translocates into plant cells and inhibits immunity.
DOI: https://doi.org/10.7554/eLife.34902.007
**Figure supplement 2.** VdSCP41 inhibits pathogen-induced SA accumulation and gene expression.
DOI: https://doi.org/10.7554/eLife.34902.008

*Arabidopsis* transgenic lines expressing Δ*spVdSCP41* were constructed and assessed for PAMP-induced defense responses. Flg22 is the best-characterized PAMP derived from a bacterial pathogen, and it induces the expression of PTI-responsive genes in wildtype (WT) *Arabidopsis* plants. We observed reduced induction of flg22-induced *ICS1* in two independent *VdSCP41*-expressing lines (*Figure 2—figure supplement 1E*), suggesting inhibition of PTI conferred by VdSCP41.

NLP proteins derived from bacterial, oomycete and fungal organisms have recently been characterized as PAMPs. A conserved 20-amino-acid peptide (nlp20) within NLP represents the active immunogenic motif that induces PTI in plants (*Albert et al., 2015*; *Böhm et al., 2014*; *Ottmann et al., 2009*). We previously reported the immune-inducing activity of VdNLP1 and VdNLP2 derived from V592 in *N. b.*, *Arabidopsis*, and cotton plants (*Zhou et al., 2012*). A corresponding peptide located in VdNLP2 derived from V592 (nlp20[Vd2]) was synthesized and shown to cause upregulation of cotton *PR* genes (*Du et al., 2017*), indicating the immunogenic activity of this peptide in cotton. Treatment of *Arabidopsis* with nlp20[Vd2] significantly induced *ICS1* and *FMO1* expression (*Figure 2C–D*, *Figure 2—source data 1*), indicating that nlp20[Vd2] also exhibits the immunogenic activity characteristic of a PAMP in *Arabidopsis*. In transgenic lines expressing *VdSCP41*, but not transgenic lines expressing NLS mutated *VdSCP41* (Δ*spVdSCP41*[-nls]), suppression of nlp20[Vd2]-induced *ICS1* (*Figure 2C*) and *FMO1* (*Figure 2D*) was observed, suggesting that VdSCP41 expression in plants inhibits nlp20[Vd2]-triggered immunity. *ICS1* encodes the key enzyme controlling SA production and is required for pathogen-induced SA accumulation (*Wildermuth et al., 2001*). We next quantified SA production in response to a nonpathogenic bacterial pathogen, *Pst* DC3000 *hrcC⁻*, in both WT and Δ*spVdSCP41*-expressing plants. Consistent with the suppression of PAMP-induced *ICS1* expression, transgenic lines expressing Δ*spVdSCP41* accumulated less free SA in response to *Pst hrcC⁻* than did WT plants (*Figure 2—figure supplement 2A*).

Inoculation of *Arabidopsis* with the WT *V. dahliae* strain V592 induced the expression of *ICS1* and *FMO1* (*Figure 2—figure supplement 2B–C*). This induced expression was further enhanced when the plants were inoculated with the VdΔ*scp41* mutant instead of V592 (*Figure 2—figure supplement 2B–C*), indicating that VdSCP41 suppresses the induced expression of *ICS1* and *FMO1* during *V. dahliae* infection. Taken together, these results reveal an inhibitory role of VdSCP41 in modulating plant immunity.

## *Arabidopsis* master immune regulators CBP60g and SARD1 are targeted by VdSCP41

Modulating the activity of plant immune components is a strategy commonly used by effectors to suppress host immunity. To explore the virulence mechanisms employed by VdSCP41 in inhibiting plant immunity, we next searched for plant components that are targeted by VdSCP41. Δ*spVdSCP41* was fused with a 3 × FLAG tag and transiently expressed in *Arabidopsis* protoplasts, before the VdSCP41-containing protein complexes were purified. Protein lysates were immuno-

precipitated using anti-FLAG-conjugated beads and subjected to tandem mass spectrometry. The plant CBP CBP60g was identified as a candidate interactor of VdSCP41 (*Supplementary file 1*).

CBP60g was then fused with a 3 × HA tag and used for reverse co-immunoprecipitation (Co-IP) analysis to verify its interaction with VdSCP41. FLAG-tagged VdSCP41 was transfected, either alone or together with HA-tagged CBP60g, into *Arabidopsis* protoplasts for transient expression. Anti-HA IP followed by an anti-FLAG immunoblot revealed that VdSCP41 was co-purified with CBP60g from plant cells (*Figure 3—figure supplement 1A*). VdSCP41 was further divided into an N-terminal portion (VdSCP41N, amino acids 1–213) and a C-terminal portion (VdSCP41C, amino acids 163-end), which exhibited an overlap of 50 amino acids, which were used to test interactions with CBP60g. Co-IP analysis revealed that VdSCP41C is sufficient for interaction with CBP60g (*Figure 3A*).

Quantitative luciferase complementation imaging assays were performed to further verify the interaction between VdSCP41 and CBP60g in *N. b.*. Co-expression of the N-terminal region of luciferase (NLuc)-tagged BIK1 and the C-terminal region of luciferase (CLuc)-tagged XLG2 driven by the35S promoter was performed as a positive interaction control (*Liang et al., 2016*). The co-expression of NLuc-VdSCP41 and CLuc-CBP60g driven by the 35S promoter in *N. b.* resulted in much higher luciferase activity than did the co-expression of NLuc-VdSCP41 and CLuc-XLG2 or of CLuc-CBP60g and NLuc-BIK1 (*Figure 3B–C*, *Figure 3—source data 1*), confirming the interaction between VdSCP41 and CBP60g in the plants. The expression levels of the NLuc- and CLuc-fusion proteins were further detected by immunoblotting (*Figure 3—figure supplement 1B*).

The nuclear localization of VdSCP41 prompted us to examine whether CBP60g co-localizes with VdSCP41 in the nucleus. GFP-tagged CBP60g mainly localized in the nucleus when it was expressed alone in *N. b.* cells (*Figure 3—figure supplement 1C*). GFP-tagged CBP60g was co-expressed with mCherry-tagged VdSCP41 without signal peptide (ΔspVdSCP41-mCherry) in *N. b.* leaves through *Agrobacterium*-mediated transient expression. An overlay of the results of GFP and mCherry fluorescence imaging indicated co-localization of ΔspVdSCP41 and CBP60g in the nucleus of *N. b.* cells (*Figure 3D*). The protein expression level of GFP-CBP60g, ΔspVdSCP41-mCherry and ΔspVdSCP41$_{nls}$-mCherry was detected by immunoblot with the indicated antibodies (*Figure 3—figure supplement 1D*). In addition to co-localization, co-expression of ΔspVdSCP41-mCherry significantly increased the nuclear accumulation of CBP60g-GFP (*Figure 3D*), which was not observed when ΔspVdSCP41$_{-nls}$–mCherry with a mutated NLS was co-expressed with CBP60g-GFP (*Figure 3D*).

CBP60g was induced by pathogen and PAMP treatments and was required for full resistance against bacterial pathogens (*Wang et al., 2011*; *Zhang et al., 2010*). A closely related protein in the CBP family, SARD1, functions partially redundantly with CBP60g in bacterial resistance (*Sun et al., 2015*; *Wang et al., 2011*). We also detected an interaction between VdSCP41 and SARD1 by Co-IP in *Arabidopsis* protoplasts (*Figure 3—figure supplement 2A*). VdSCP41 co-localized with SARD1 in *N. b.* leaves and increased its nuclear accumulation (*Figure 3—figure supplement 2B*), indicating similar targeting of SARD1 by VdSCP41 in *N. b.*. It is unlikely to be an interaction between CBP60g and SARD1 because the luciferase complementation assay did not show interaction between CLuc-tagged CBP60g and NLuc-tagged SARD1 (*Figure 3—figure supplement 3*). Taken together, the results described above demonstrated that both CBP60g and SARD1 are targeted by VdSCP41.

## VdSCP41 interferes with the transcription factor activity of CBP60g

CBP60g encodes a plant-specific transcription factor that regulates the expression of a number of defense-related genes. The fact that CBP60g functions as a master transcription regulator (*Sun et al., 2015*; *Wang et al., 2011*) prompted us to examine whether the targeting of CBP60g by VdSCP41 affects the induction of its target genes. Dual reporter analyses revealed that *CBP60g* expression in *Arabidopsis* protoplasts significantly enhanced the expression of *ICS1::LUC* or *FMO1::LUC* (firefly luciferase) (*Figure 4A–B*, *Figure 4—source data 1*). Co-expression of *VdSCP41* inhibited CBP60g-induced *ICS1::LUC* (*Figure 4A*) and *FMO1::LUC* (*Figure 4B*) expression, whereas the co-expression of *VdSCP41N*, which is unable to bind CBP60g, was impaired in this inhibition (*Figure 4A–B*). CBP60g was next divided into an N-terminal portion (CBP60gN, amino acids 1–361) containing its DNA-binding domain and a C-terminal portion (CBP60gC, amino acids 211-end) lacking the functional DNA-binding domain (*Zhang et al., 2010*), which exhibited an overlap of 150 amino acids. CBP60gN and CBP60gC were then used to test interactions with VdSCP41. Co-IP

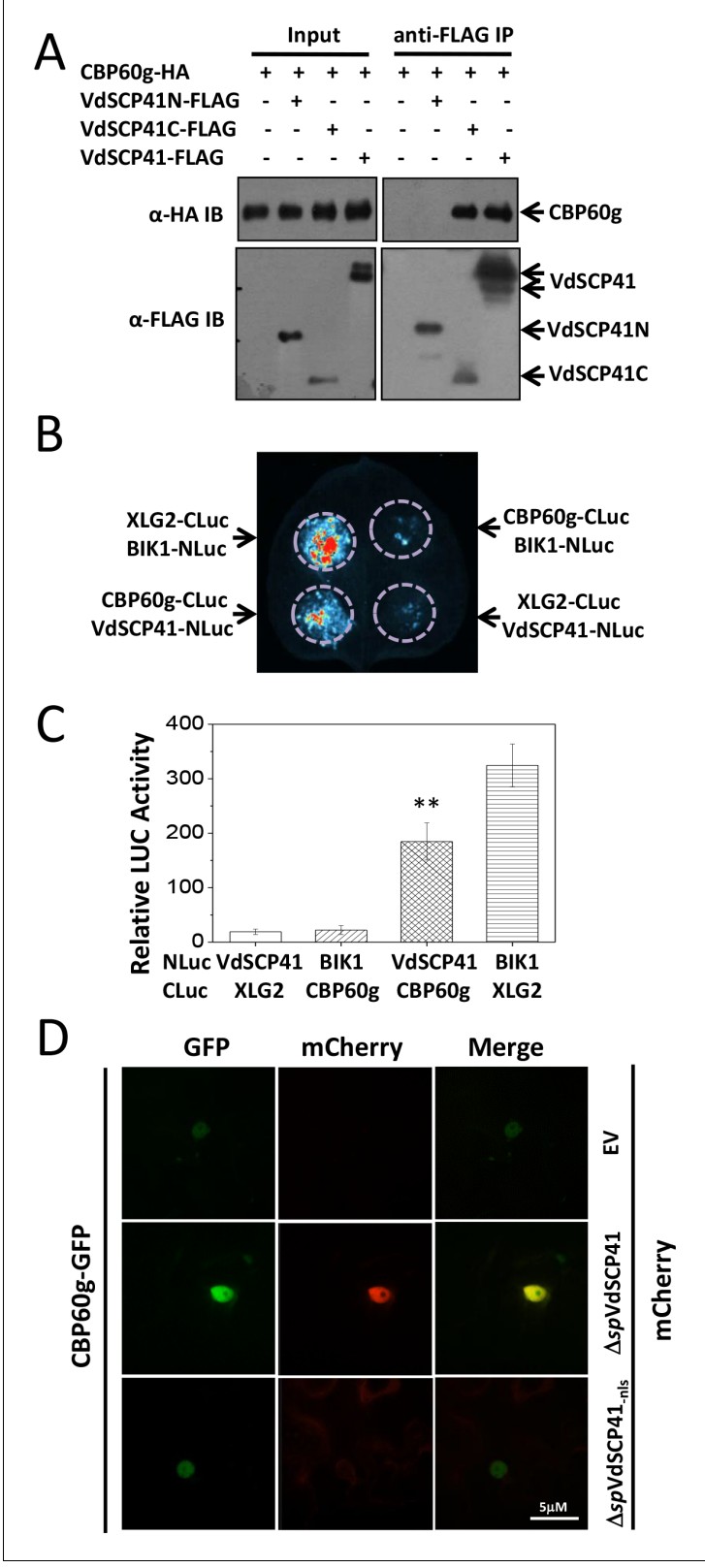

**Figure 3.** VdSCP41 interacts with CBP60g in plants. (**A**) VdSCP41C interacts with CBP60g. *Arabidopsis* protoplasts were transfected with the indicated constructs. Protein was extracted 16 hr post transfection and immunoprecipitated with anti-FLAG. The presence of VdSCP41-FLAG or CBP60g-HA in the purified complex was detected by anti-FLAG or anti-HA immunoblot as indicated. (**B**) VdSCP41 interacts with CBP60g in *N. b.*.

*Figure 3 continued on next page*

*Figure 3 continued*

Luciferase imaging of VdSCP41 and CBP60g interaction in *N. b.* leaves. *N. b.* leaves infiltrated with an *Agrobacterium* strain carrying constructs as indicated were subjected to luciferase complementation imaging assay. (C) Quantitative luminescence of VdSCP41 and CBP60g interaction. *N. b.* leaves infiltrated with the indicated constructs were sliced into strips, and their relative luminescence was determined using a microplate luminometer. Error bars indicate standard deviations of three technical repeats. ** Indicates significant difference at *P*-value < 0.01. The experiments were repeated three times with similar results. (D) VdSCP41 co-localizes with CBP60g and increases its nuclear accumulation. Representative confocal images of CBP60g-GFP subcellular accumulation were visualized using a spin-disk microscope. An *Agrobacterium* strain carrying CBP60g-GFP was *Agro*-infiltrated into *N. b.* leaves alone, or together with an *Agrobacterium* strain carrying Δ*sp*VdSCP41-mCherry, Δ*sp*VdSCP41$_{-nls}$-mCherry. An overlay of GFP and mCherry fluorescence imaging was visualized 48 hr post *Agro*-infiltration in *N. b.* leaves. The experiments were repeated three times with similar results.

DOI: https://doi.org/10.7554/eLife.34902.010

The following source data and figure supplements are available for figure 3:

**Source data 1.** Source data for *Figure 3*.
DOI: https://doi.org/10.7554/eLife.34902.014
**Figure supplement 1.** VdSCP41 interacts with CBP60g.
DOI: https://doi.org/10.7554/eLife.34902.011
**Figure supplement 2.** VdSCP41 interacts and co-localizes with SARD1.
DOI: https://doi.org/10.7554/eLife.34902.012
**Figure supplement 3.** Luciferase complementation assay did not reveal an interaction between CBP60g and SARD1.
DOI: https://doi.org/10.7554/eLife.34902.013

analysis showed that VdSCP41 binds to CBP60gC, but not CBP60gN (*Figure 4C*). The results indicated that VdSCP41 binds the C-terminal portion of CBP60g to interfere with its activity.

To test whether VdSCP41 targeting directly affects the DNA-binding activity of CBP60g, recombinant GST-tagged CBP60g and His-tagged VdSCP41C were purified and used for electrophoretic mobility shift assays (EMSAs). GST-CBP60g showed a specific binding to a 60-bp DNA fragment (*ICS1* promoter probe) within the *ICS1* promoter, which is reduced by the addition of unlabelled probe, as previously reported (*Zhang et al., 2010*) (*Figure 4D*). The preincubation of VdSCP41C with CBP60g significantly reduced the DNA-binding activity of CBP60g, whereas a soluble fragment of His-tagged VdSCP41 which does not contain the C-terminal portion (VdSCP41$_{100-163}$) did not (*Figure 4D*). Another His-tagged *V. dahliae* protein (VDAG_01962) also did not affect the DNA-binding activity of CBP60g (*Figure 4D*). Coexpression of VdSCP41 did not lead to cleavage or mobility shift of CBP60g (*Figure 4—figure supplement 1*), suggesting that VdSCP41 is unlikely to act as a protease to target CBP60g. The results proved that binding of VdSCP41C to CBP60g directly inhibits the DNA-binding activity of CBP60g.

## CBP60gC harbors a transcription activation domain that is required for interaction with VdSCP41

The binding of VdSCP41 to the C-terminal portion of CBP60g prompted us to test the role of CBP60gC in CBP60g-mediated gene activation. Compared to the full induction of *ICS1::LUC* or *FMO1::LUC* by CBP60g (*Figure 5A*, *Figure 5—source data 1*), the deletion of the C-terminal portion (ΔC-CBP60g) dramatically compromised its activity to induce both *ICS1::LUC* and *FMO1:: LUC* (*Figure 5A*), indicating that CBP60gC is required for CBP60g-mediated gene activation. The results suggest that putative transcription activator activity may be harbored within the CBP60gC. The basic helix-loop-helix (bHLH) transcription factor MYC2 directly binds to the G-box-like (CANNTG) element (*Dombrecht et al., 2007*; *Godoy et al., 2011*; *Lian et al., 2017*) via its bHLH domain to regulate the expression of its target genes, such as the *TERPENE SYNTHASE gene 10* (*TPS10*) (*Li et al., 2014b*). We therefore took advantage of bHLH-mediated binding to the *TPS10* promoter to examine the putative transcription activator activity of CBP60gC. A fragment within the C-terminal portion of CBP60g (amino acids 211–440) activated *TPS10::LUC* reporter when it was fused with the bHLH domain of MYC2 (bHLH-CBP60g$_{211-440}$) (*Figure 5B*, *Figure 5—source data 1*) rather than bHLH$_{MYC2}$ alone, indicating that the CBP60g$_{211-440}$ harbors transcription activator

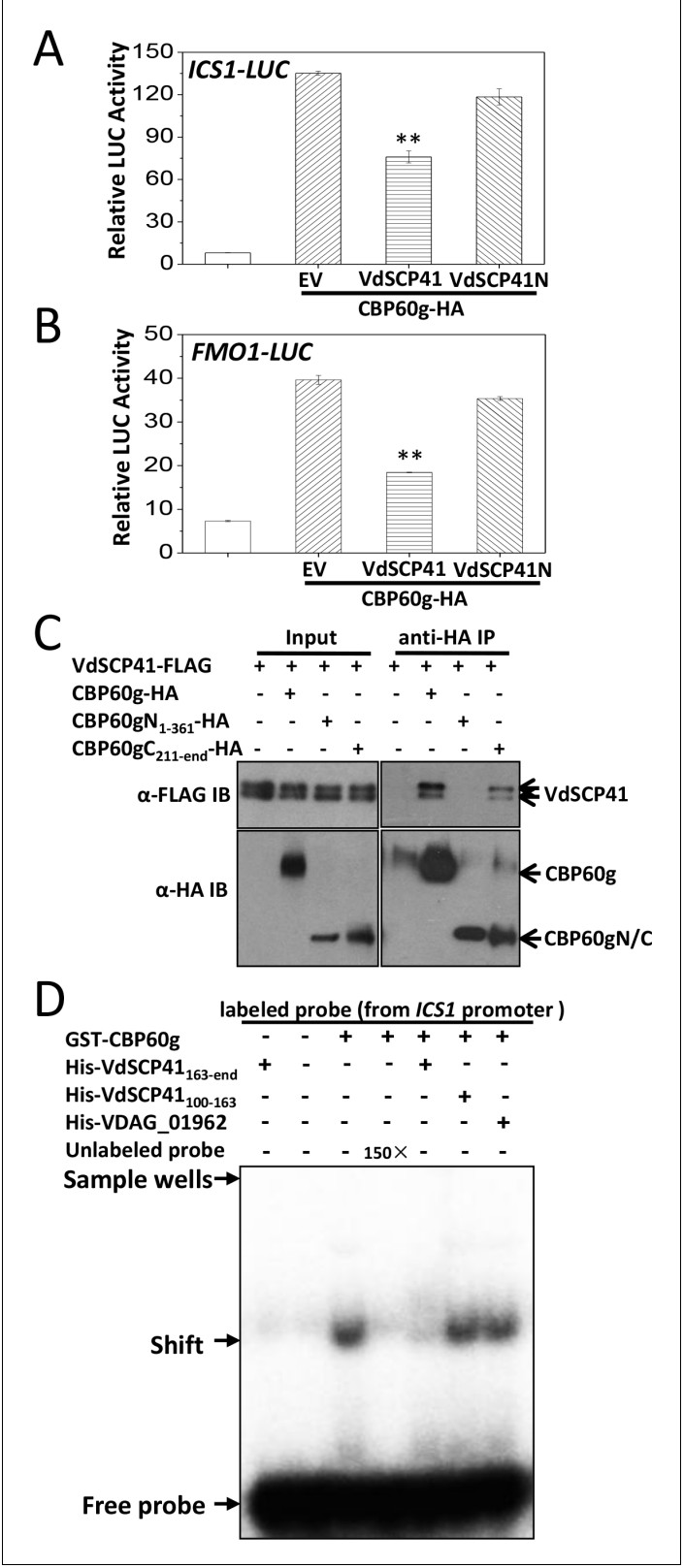

**Figure 4.** VdSCP41 binds the C-terminal portion of CBP60g to interfere with its activity. (**A–B**) Expression of *VdSCP41* inhibits the CBP60g-activated *ICS1::LUC* (**A**) and *FMO1::LUC* (**B**) reporter. *ICS1::LUC* or *FMO1::LUC* were transfected alone, or together with CBP60g and VdSCP41 or its variants, as indicated. *35S::RLUC* was co-transfected as internal control. Error bars indicate standard deviations of three technical repeats. ** Indicates

*Figure 4 continued on next page*

*Figure 4 continued*

significant difference between VdSCP41 and EV (empty vector) at a *P*-value of < 0.01. (**C**) CBP60gC interacts with VdSCP41. *Arabidopsis* protoplasts were transfected with VdSCP41-FLAG alone or together with CBP60gN-HA, CBP60gC-HA, or CBP60g-HA. Protein was extracted 16 hr post transfection and immunoprecipitated with anti-HA antibody. The presence of VdSCP41-FLAG in the purified complex was detected by anti-FLAG immunoblot. The experiments were repeated three times with similar results. (**D**) VdSCP41 inhibits the DNA-binding activity of CBP60g. GST-CBP60g was incubated with $[\gamma-32P]$ATP-labeled 60-bp double-stranded DNA probe within the *ICS1* promoter, and subjected to electrophoretic mobility shift assay (EMSA). Unlabeled probe was used as competitors (150×) for binding. His-tagged VdSCP41C, VdSCP41$_{100-163}$ or VDAG_01962 was preincubated with GST-CBP60g for 30 min at room temperature where needed (as indicated) before EMSA. The experiments were repeated three times with similar results.

DOI: https://doi.org/10.7554/eLife.34902.015

The following source data and figure supplement are available for figure 4:

**Source data 1.** Source data for *Figure 4*.
DOI: https://doi.org/10.7554/eLife.34902.017
**Figure supplement 1.** VdSCP41 does not cleave CBP60g.
DOI: https://doi.org/10.7554/eLife.34902.016

activity. Moreover, Co-IP analysis showed that CBP60g$_{211-end}$ but not CBP60g$_{441-end}$ co-purified with VdSCP41, indicating the requirement for CBP60g$_{211-440}$ for binding to VdSCP41 (*Figure 5C*). Thus, CBP60gC harbors a transcription activation domain, CBP60g$_{211-440}$, that is required for VdSCP41 targeting.

The ΔC-CBP60g was further co-transfected with CBP60g, together with *ICS1::LUC* or *FMO1::LUC*. Dual reporter analyses indicated that co-expression of *ΔC-CBP60g* significantly suppressed CBP60g-induced *ICS1::LUC* and *FMO1::LUC* (*Figure 5A*), suggesting a dominant-negative effect of ΔC-CBP60g on the activity of CBP60g.

## *CBP60g* and *SARD1* contribute to *Arabidopsis* resistance against *V. dahliae*

*SARD1* functions both redundantly and differentially with *CBP60g* (*Sun et al., 2015*; *Wang et al., 2011*). The finding that VdSCP41 targeted both CBP60g and SARD1 prompted us to assess the roles of *CBP60g* and *SARD1* in resistance to *V. dahliae*. The WT and *cbp60g-1/sard1-1* double mutant plants were inoculated with V592 for the assessment of disease symptoms. As shown in *Figure 6*, the *cbp60g-1/sard1-1* double mutant displayed compromised resistance compared with the WT plants, demonstrating a contribution of *CBP60g* and *SARD1* to *V. dahliae* resistance.

We next investigated the requirement for *CBP60g* and *SARD1* for VdSCP41-mediated virulence. The WT and *cbp60g-1/sard1-1* double mutant plants were inoculated with the VdΔ*scp41* mutant. When compared with V592, the VdΔ*scp41* mutant displayed reduced virulence on the WT plants. The reduced virulence arising from *VdSCP41* deletion was partially impaired in the *cbp60g-1/sard1-1* double mutant plants compared with that in the WT plants (*Figure 6*, *Figure 6—source data 1*). The results indicated that both *CBP60g* and *SARD1* are required for full virulence conferred by VdSCP41. However, we still observed reduced virulence of the VdΔ*scp41* mutant compared with that of V592 on the *cbp60g-1/sard1-1* double mutant plants, suggesting the existence of additional targets for VdSCP41 during *V. dahliae* infection.

## VdSCP41 interacts with and co-localizes with GhCBP60b

As in *Arabidopsis*, we observed similar compromised virulence of the VdΔ*scp41* mutant in upland cotton (*Figure 1E*) compared with V592. The results prompted us to test the putative VdSCP41 targeting of GhCBP60b, the closest protein homolog of CBP60g and SARD1 in cotton. Co-IP analysis revealed an interaction between VdSCP41 and GhCBP60b (*Figure 7A*). To examine the subcellular localization of GhCBP60b, GhCBP60b-GFP was constructed for transient expression in *N. b.* leaves. GhCBP60b-GFP localized to the nucleus of *N. b.* cells, and co-expression of GhCBP60b-GFP with VdSCP41-mCherry revealed co-localization of VdSCP41 and GhCBP60b in the nucleus of *N. b.* cells (*Figure 7B*), suggesting conserved targeting of *Arabidopsis* CBP60g and SARD1 and of cotton GhCBP60b by VdSCP41.

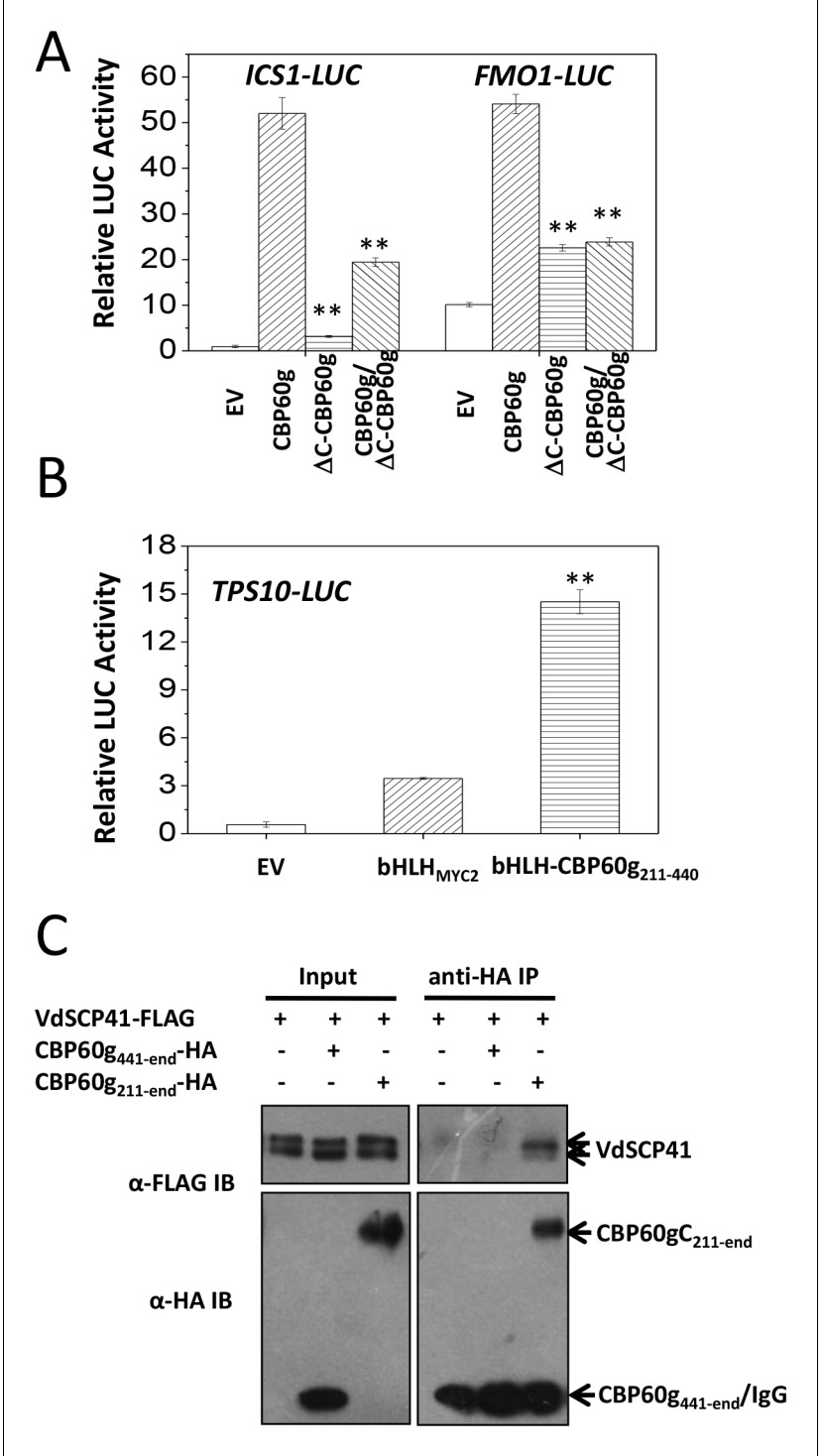

**Figure 5.** CBP60gC harbors a transcription activation domain that is required for interaction with VdSCP41. (**A**) CBP60gC is required for CBP60g-mediated *ICS1* and *FMO1* activation. *ICS1::LUC* or *FMO1::LUC* was transfected alone, or together with CBP60g or its variants, as indicated. ** Indicates significant difference between ΔC-CBP60g or ΔC-CBP60g/CBP60g and CBP60g at a *P*-value of < 0.01. (**B**) CBP60g$_{211-440}$ harbors transcription activator activity. *TPS10::LUC* was transfected alone, or together with bHLH$_{MYC2}$ or bHLH-CBP60g$_{211-440}$, as indicated. *35S:: RLUC* was co-transfected as internal control. ** Indicates significant difference between bHLH-CBP60g$_{211-440}$ and bHLH$_{MYC2}$ at a *P*-value of < 0.01. *LUC* reporter activity was determined 16 hr post transfection. The experiments were repeated three times with similar results. (**C**) CBP60g$_{211-440}$ is required for interacting with VdSCP41. *Arabidopsis* protoplasts were transfected with the indicated constructs. Protein was extracted 16 hr post

*Figure 5 continued*

transfection and immunoprecipitated with anti-HA. The presence of VdSCP41-FLAG or CBP60g variants in the purified complex was detected by anti-FLAG or anti-HA immunoblot as indicated.

DOI: https://doi.org/10.7554/eLife.34902.018

The following source data is available for figure 5:

**Source data 1.** Source data for *Figure 5*.

DOI: https://doi.org/10.7554/eLife.34902.019

## *GhCBP60b* is required for cotton resistance against *V. dahliae*

To further examine whether *GhCBP60b* functions in resistance to *V. dahliae* in cotton, we generated a virus-induced gene silencing (VIGS) vector (*Liu et al., 2002*) targeting *GhCBP60b* (pTRV2-*GhCBP60b*). Cotton plants were infiltrated with pTRV1 together with pTRV2 or pTRV2-*GhCBP60b* and further inoculated with *V. dahliae*. When compared with pTRV2, pTRV2-*GhCBP60b*-infiltrated cotton plants exhibited higher ratios of wilting (*Figure 7C*, *Figure 7—source data 1*) and more severe disease symptoms (*Figure 7D–E*, *Figure 7—source data 1*), indicating a role for *GhCBP60b* in cotton resistance to *V. dahliae*. The reduced expression of *GhCBP60b* in pTRV2-*GhCBP60b*-infiltrated plants was verified using RT-PCR (*Figure 7F*, *Figure 7—source data 1*). The results support the targeting of GhCBP60b by VdSCP41 for virulence during *V. dahliae* infection in cotton.

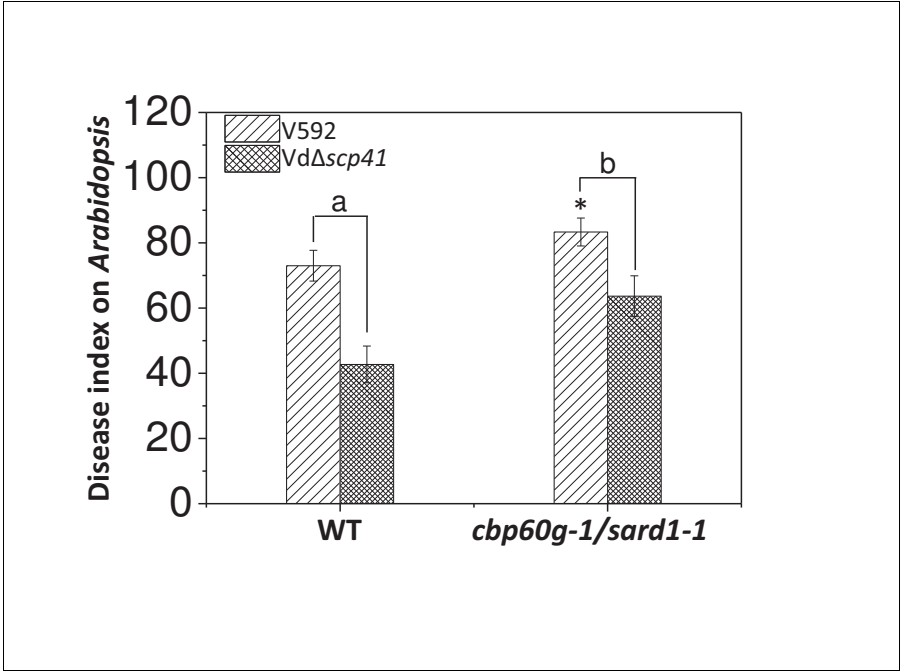

**Figure 6.** *CBP60g* and *SARD1* are required for VdSCP41-mediated virulence. Wildtype (WT) and *cbp60g-1/sard1-1* double mutant plants were inoculated with V592 and VdΔ*scp41* mutant strains. The plants were subjected to disease index analyses 3–4 weeks post inoculation. The disease indexes were evaluated with three replicates generated from 24 plants for each inoculum. Error bars indicate standard deviations of three biological replicates. *Indicates significant difference of V592 virulence at a *P*-value of < 0.05 between WT and *cbp60g-1/sard1-1* double mutant plants. Student's t-test was carried out to determine the significance of difference between indexes collected from three biological replicates. Lower case letters indicate a significant difference at a *P*-value of < 0.05. The experiments were repeated three times with similar results.

DOI: https://doi.org/10.7554/eLife.34902.020

The following source data is available for figure 6:

**Source data 1.** Source data for *Figure 6*.

DOI: https://doi.org/10.7554/eLife.34902.021

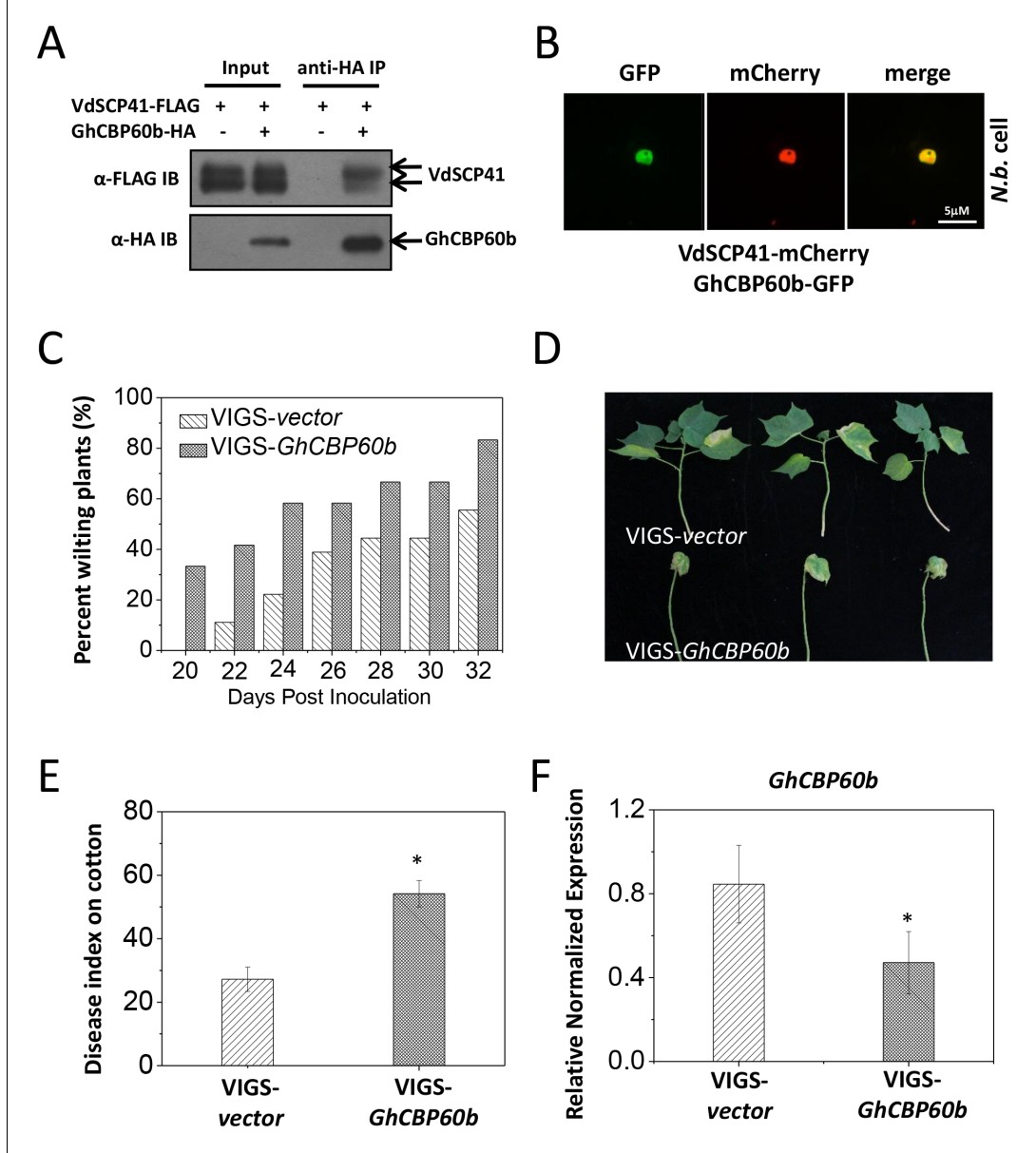

**Figure 7.** *GhCBP60b* is targeted by VdSCP41 and required for cotton resistance against *V. dahliae*. (**A**) VdSCP41 is co-purified with GhCBP60b in *Arabidopsis* protoplasts. *Arabidopsis* protoplasts were transfected with VdSCP41-FLAG alone or together with GhCBP60b-HA. Protein was extracted 16 hr post transfection and immunoprecipitated with anti-HA. The presence of VdSCP41-FLAG in the purified complex was detected by anti-FLAG immunoblot. (**B**) VdSCP41 co-localizes with GhCBP60b. GFP-tagged GhCBP60b and mCherry-tagged VdSCP41 were co-expressed in *N. b.* leaves. An overlay of GFP and mCherry fluorescence imaging was visualized 48 hr post *Agro*-infiltration. The experiments were repeated three times with similar results. (**C**) Cotton VIGS-*GhCBP60b* plants develop symptoms more rapidly than VIGS-*vector* plants. The percentage of plants showing the *Verticillium* wilt phenotype at the indicated time after infection is shown. The disease ratio was scored using 15 plants per treatment and the assays were repeated three times with similar results. (**D–E**) *GhCBP60b* is required for full resistance against *V. dahliae*. Cotton seedlings were infiltrated with *Agrobacterium* carrying pTRV1 together with pTRV2 or pTRV2-*GhCBP60b* as indicated. *Verticillium dahliae* strain V592 was inoculated 10 days post *Agrobacterium* infiltration. Plants showing disease symptoms were photographed 30 days post *V. dahliae* inoculation (**D**). Disease index analyses of VIGS-*vector* or VIGS-*GhCBP60b* plants infected with V592 (**E**). The plants were subjected to disease index analyses 3–4 weeks post inoculation. The disease indexes were evaluated in three replicates generated from 24 cotton plants for each inoculum. The experiments were repeated three times with similar results. Error bars indicate standard deviations of three technical repeats within one biological experiment. The Student's t-test was carried out to determine the significance of difference. *Indicates significant difference at a *P*-value of < 0.05. (**F**) Expression of *GhCBP60b* is reduced in plants that are infiltrated with *Agrobacterium* carrying pTRV1 together with pTRV2-*GhCBP60b* (VIGS-*GhCBP60b*). Total RNA from infiltrated plants was extracted for RT-PCR analyses of *GhCBP60b* expression 20 days post *V. dahliae* inoculation. Error bars indicate standard deviations. *Indicates significant difference at a *P*-value of < 0.05.

*Figure 7 continued on next page*

*Figure 7 continued*

DOI: https://doi.org/10.7554/eLife.34902.022

The following source data is available for figure 7:

**Source data 1.** Source data for *Figure 7*.

DOI: https://doi.org/10.7554/eLife.34902.023

## Discussion

In this study, we identified VdSCP41 as an intracellular effector that is crucial for *V. dahliae* virulence. VdSCP41 targets *Arabidopsis* CBP60g and SARD1 to interfere with the induction of their target genes. CBP60g and SARD1 are required for *Arabidopsis* resistance to *V. dahliae* and for VdSCP41-mediated virulence. Silencing of *GhCBP60b* compromised cotton resistance to *V. dahliae*. The results revealed a conserved and important role for *Arabidopsis* CBP60g and SARD1 and for cotton GhCBP60b, which are modulated by an intracellular effector, in plant resistance against *V. dahliae*.

### VdSCP41 is secreted by *V. dahliae* and translocates into plant cells

To suppress plant defense actively or to modulate host physiology to benefit pathogenic fitness, oomycete and fungal pathogens deliver hundreds of effectors through specialized intracellular fungal structures, such as haustoria and infection hyphaea, into the host apoplastic space or directly into plant cells (*Lo Presti et al., 2015*). Although haustoria have not been observed during infection, *V. dahliae* develops a penetration peg from a hyphopodium when infecting plant roots (*Zhao et al., 2016*). During *V. dahliae* infection, the penetration peg that developed from the hyphopodium further develops a hyphal neck and forms a septin ring that partitions the hyphopodium and invasive hyphae. This septin-organized apparatus functions as a fungus–host interface for the dynamic delivery of secretory proteins, such as SCPs (*Zhao et al., 2016Zhou et al., 2017*).

To date, relatively few *V. dahliae*-secreted effectors have been characterized as functioning inside host cells to modulate host immunity. Here, we provided evidence of the secretion and translocation of VdSCP41, which is secreted by *V. dahliae*, into plant cells. VdSCP41 contains a signal peptide and localizes to the base of the hyphopodium to form a septin-like ring during infection (*Figure 2A*), indicating that it is secreted via the septin-organized apparatus at the fungus–host interface. The localization of the VdSCP41 delivered by *V. dahliae* at the nucleus of onion epidermal cells (*Figure 2—figure supplement 1B*) suggested the translocation of VdSCP41 into plant cells. A few conserved motifs have been indicated to serve as signals for the uptake of fungal or oomycete effectors into plant cells. For example, a few effectors secreted by cereal powdery mildew and rust pathogens possess a Y/F/WxC motif that serves as a signal for translocation into the plant cell (*Godfrey et al., 2010*; *Spanu et al., 2010*). A set of oomycete effectors contain a RXLR-dEER motif that appears to assist in targeting effectors into plant cells (*Dou et al., 2008*; *Whisson et al., 2007*). Another set of oomycete effectors possess a FLAK motif for translocation (*Schornack et al., 2010*). However, VdSCP41 lacks any of the above conserved motifs, and the mechanisms that assist the uptake of VdSCP41 into plant cells remain unclear.

### VdSCP41 targets CBP60g and SARD1 and interferes with plant defense

CBP60g and SARD1 are master regulators in immunity. ChIP-seq analyses have allowed the identification of a large number of target genes for CBP60g and SARD1 and support a model in which CBP60g and SARD1 accumulate in the plant nucleus and act as master regulators, which activate both positive and negative immune regulators (*Sun et al., 2015*; *Wang et al., 2009*; *Zhang et al., 2010*). Genetic evidence revealed that CBP60g and SARD1 are positive immune regulators required for immune responses and bacterial resistance (*Wang et al., 2009*; *Zhang et al., 2010*). We demonstrated that VdSCP41 targets CBP60g and SARD1 and interferes with their activity, thus supporting the virulence function of VdSCP41 for immune suppression. Consistent with the inhibition of CBP60g and SARD1 activity, we observed less pathogen-induced SA accumulation in transgenic lines that expressed *VdSCP41* than in WT plants (*Figure 2—figure supplement 2A*). As *CBP60g* and *SARD1* also bind directly to the promoters of both *ALD1* and *SARD4* (*Sun et al., 2018*) (two major genes encoding pipecolic acid (Pip) biosynthesis enzymes) to regulate Pip biosynthesis, a potential contribution of Pip to *V. dahliae* resistance will be of interest for further investigation.

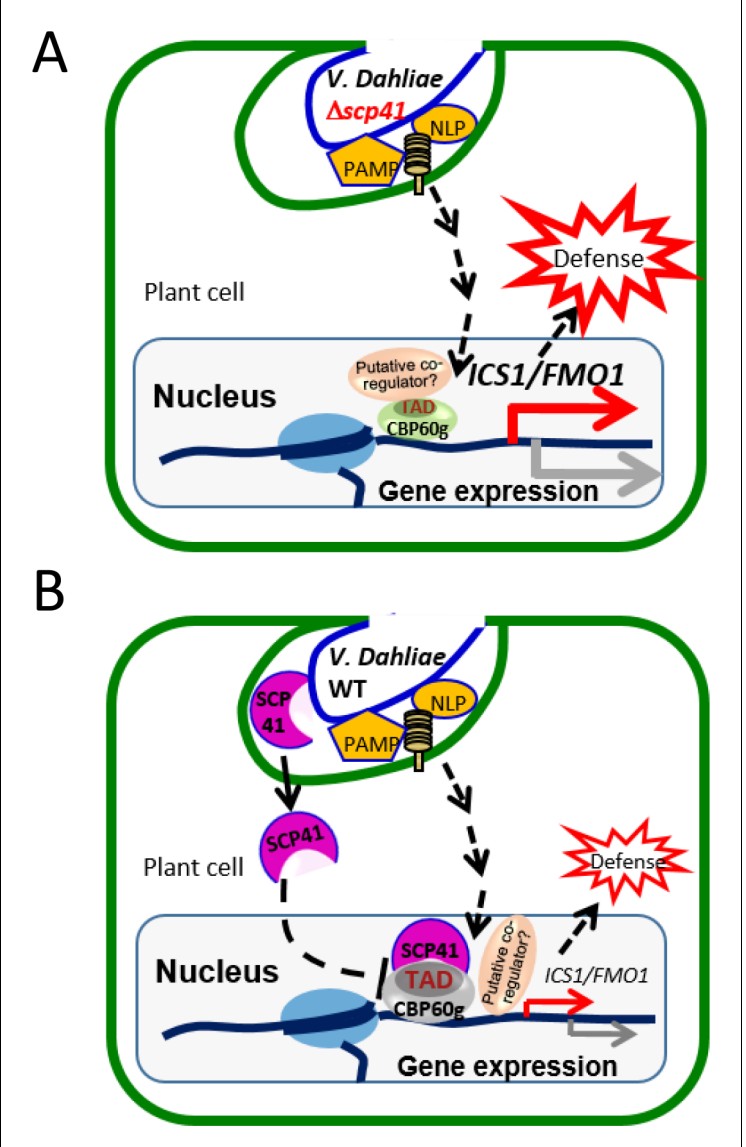

**Figure 8.** Model for VdSCP41–mediated suppression of defense in *Arabidopsis* during *V. dahliae* infection. (**A**) In the absence of VdSCP41, PAMPs derived from *V. dahliae*, such as NLPs and chitins, induce the expression of *CBP60g*, which in turn upregulates the expression of a number of immune regulators to activate defense. (**B**) In the presence of VdSCP41, VdSCP41 secreted from *V. dahliae* translocates into the nucleus of plant cells. VdSCP41 targets the transcription activation domain (TAD) of CBP60g, interrupting either the activity of this domain or the recruitment of associated co-activators via this domain, to interfere with their activity and with plant immunity against *V. dahliae*. VdSCP41-mediated over-accumulation could provide an additional strategy to further interfere with the transcription factor activity of CBP60g.
DOI: https://doi.org/10.7554/eLife.34902.024

We showed that VdSCP41 binds the C-terminal portion of CBP60g to interfere with its transcription factor activity (*Figure 4A–B*). Coexpression of VdSCP41 did not lead to cleavage or mobility shift of CBP60g (*Figure 4—figure supplement 1*), suggesting that VdSCP41 is unlikely to act as a protease to target CBP60g. In addition, CBP60g$_{211-440}$, a domain within the C-terminal portion of CBP60g, harbors transcription activator activity (*Figure 5B*) and is required for interaction with VdSCP41 (*Figure 5C*). It is likely that CBP60gC functions to promote transcriptional activation by recruiting additional activators, and that binding of VdSCP41 interrupts either the activity of this domain or the recruitment of associated activators via this domain. The dominant-negative function

of ΔC-CBP60g (*Figure 5A*) supports a dominant-negative effect on CBP60g activity arising from increased nuclear accumulation of VdSCP41-impaired CBP60g. Thus, VdSCP41-mediated over-accumulation of CBP60g provides an additional strategy to further interfere with CBP60g activity as a transcription factor. The results suggest a novel virulence strategy in which a pathogenic effector directly targets host transcription factors to interfere with their activity and to modulate plant immunity. However, it is equally possible that the increased nuclear accumulation of CBP60g results from feed-back regulation as a result of impaired CBP60g function.

## VdSCP41 functions to suppress immunity

CBP60g and SARD1 are key components of the SA signaling pathway, which serves as an attractive target for bacterial and fungal effectors (*DebRoy et al., 2004*; *Djamei et al., 2011*; *Nomura et al., 2011*). Targeting of CBP60g and SARD1 by VdSCP41 may therefore interfere with plant resistance to *V. dahliae* by manipulating SA signaling. On the other hand, ChIP-seq analyses identified a number of SA-independent regulators that are directly targeted by CBP60g and SARD1 (*Sun et al., 2015*), revealing broader and SA-independent functions of CBP60g and SARD1 in immunity. It is likely that signaling pathways other than SA are also targeted by VdSCP41 through CBP60g and SARD1 to interfere with plant immunity. Consistent with this assumption, we showed that nlp20$^{Vd2}$ functions to trigger the induction of both SA-dependent and SA-independent defense genes during *V. dahliae* infection (*Figure 2C–D*). Furthermore, VdSCP41 expression suppressed the induction of both SA-dependent *ICS1* and SA-independent *FMO1* by nlp20$^{Vd2}$ (*Figure 2C–D*). Modulation of CBP60g and SARD1 may result in interference with their function as master transcription factors in PTI and in other defense pathways.

Taken together, our results support a model in which PAMPs from *V. dahliae*, such as NLPs and chitins, are recognized by plants to induce the expression of *CBP60g* and *SARD1*, which subsequently regulate the expression of a number of immune regulators to defend against pathogen infection (*Figure 8A*). VdSCP41 secreted by *V. dahliae* functions as an intracellular effector that targets the transcription activator domain of CBPs, interrupting the function of this domain, to interfere with their transcription factor activity, and thus modulates both SA-dependent and SA-independent regulators to inhibit plant immunity against *V. dahliae* (*Figure 8B*).

# Materials and methods

## Key resources table

| Reagent type (species) or resource | Designation | Source or reference | Identifiers | Additional information |
|---|---|---|---|---|
| Gene (*Verticillium dahliae*) | *VdSCP41* | PMID:21829347 | NCBI Gene ID:20709665 | |
| Gene (*Gossypium hirsutum*) | *GhCBP60b* | | NCBI Gene ID:107899061 | |
| Strain, strain background (*Verticillium dahliae*) | V592 | PMID:21151869 | | |
| Other (cotton plant) | Xinluzao No. 16 | PMID:28282450 | | |
| Antibody | Anti-FALG (mouse monoclonal) | Sigma | RRID:AB_259529 | 1:10000 dilution |
| Antibody | Anti-HA (mouse monoclonal) | Roche | RRID:AB_514506 | 1:5000 dilution |
| Antibody | Anti-GFP (mouse monoclonal) | Roche | RRID:AB_390913 | 1:3000 dilution |
| Antibody | Anti-CLuc (mouse monoclonal) | Sigma | RRID:AB_439707 | 1:3000 dilution |
| Antibody | Anti-mCherry | Easybio | | 1:3000 dilution |
| Peptide | nlp20$^{Vd2}$ | PMID:28755291 | | |
| Recombinant DNA reagent | pTRV1 (VIGS vector) | PMID:12220268 | | |
| Recombinant DNA reagent | pTRV2 (VIGS vector) | PMID:12220268 | | |

## Fungal strains, plant materials and antibodies

The *Verticillium dahliae* strain V592 (*Gao et al., 2010*) was used in this study. *Verticillium dahliae* strains were grown on potato dextrose agar (PDA) medium at 25°C in the dark. To collect conidia, the mycelial plugs were cultured in potato dextrose broth (PDB) liquid medium at 25°C with shaking at 200 rpm for 3–5 days. Cotton plants ('Xinluzao No. 16') were used for virulence assessment in this

study (*Zhou et al., 2017*). *Arabidopsis thaliana* plants used in this study include Col-0 (wild-type) and the *cbp60g-1/sard1-1* mutant (*Zhang et al., 2010*). *VdSCP41* was amplified from the V592 cDNA and cloned into pCambia1300-35S-FLAG (*Zhang et al., 2010*) to construct *Arabidopsis* transgenic lines expressing *VdSCP41*. Antibodies used in this study include anti-FLAG (RRID:AB_259529), anti-HA (RRID:AB_514506), anti-CLuc (RRID:AB_439707), anti-GFP (RRID:AB_390913), and anti-mCherry (Easybio, BE2026).

## Generation of deletion and complementation fungal strains

The upstream and downstream flanking sequences were PCR amplified from V592 genomic DNA and cloned into a pGKO-HPT vector (*Wang et al., 2016*). The resulting construct was transformed into *Agrobacterium tumefaciens* EHA105, and used for *A. tumefaciens*-mediated transformation (ATMT) to generate the Vd∆*scp41* mutant strain (*Wang et al., 2016*). The genomic region of *VdSCP41*, including 1.5 kb upstream from the start codon, was amplified and cloned into a pNat-Tef-TrpC vector (*Zhou et al., 2017*) to generate a construct for complementation. The resulting construct was transformed into *Agrobacterium tumefaciens* EHA105, and used for ATMT to generate a Vd∆*scp41/VdSCP41-GFP* strain. Primers used in this study are listed in *Supplementary file 2*.

## Protein complex purification and mass spectrum analysis

*Arabidopsis* protoplasts isolated from 10 gram leaves were transfected with ∆spVdSCP41-FLAG, ∆spVdSCP45-FLAG, or empty vector for protein expression. Transfected protoplasts were collected and total protein was extracted with extraction buffer containing 50 mM HEPES (pH 7.5), 150 mM KCl, 1 mM EDTA, 1 mM DTT, 0.2% Triton X-100, and 1 × proteinase inhibitor cocktail. Total protein was incubated with 50 µl anti-FLAG agarose beads (Sigma) for 12 hr at 4°C. The immunocomplex was washed three times using the buffer described above and eluted with 100 µl 1 µg/µl 3 × FLAG peptide. The eluted proteins were run 10 mm into the separating gel and stained with Proteo Silver stain kit (Sigma). Total protein was destained and digested in-gel with sequencing grade trypsin (10 ng/mL trypsin, 50 mM ammonium bicarbonate [pH 8.0]) overnight. Peptides were sequentially extracted with 5% formic acid/50% acetonitrile and 0.1% formic acid/75% acetonitrile and concentrated to 20 µl. The extracted peptides were separated by an analytical capillary column packed with 5 mm spherical C18 reversed-phase material. The eluted peptides were sprayed into a LTQ mass spectrometer (Thermo Fisher Scientific) equipped with a nano-ESI ion source. The mass spectrometer was operated in data-dependent mode with one MS scan followed by five MS/MS scans for each cycle. Database searches were performed on an in-house Mascot server (Matrix Science Ltd., London, UK) against the IPI (International Protein Index) *Arabidopsis* protein database.

## Co-immunoprecipitation assay in protoplasts

Five-week-old *Arabidopsis* plants were used for protoplast isolation. pUC-35S-VdSCP41-FLAG, or its variant constructs, was co-transfected with pUC-35S-CBP60g-HA, or its variant constructs, pUC-35S-SARD1-HA or pUC-35S-GhCBP60b-HA, into *Arabidopsis* protoplasts. Total protein was extracted with extraction buffer. For anti-HA IP, total protein was incubated with 2 µg of anti-HA antibody together with protein A agarose at 4°C for 4 hr. The agarose beads were collected and boiled for 5 min with 1 × protein loading buffer. Immunoprecipitates were separated by 10% SDS-PAGE, and the presence of VdSCP41-FLAG or its variants, CBP60g-HA or its variants, SARD1-HA, or GhCBP60b-HA was detected by anti-FLAG or anti-HA immunoblot.

## SA measurement

SA was extracted and measured following the method described previously (*Sun et al., 2015*). Around 0.3 grams of leaf tissue, collected from 4-week-old plants, was ground into powder in liquid nitrogen. Plant leaves were infiltrated with or without *Pst* DC3000 *hrcC*⁻ (OD600 = 0.1) 12 hr before sample collection. Three samples were analysed for each treatment. The samples were extracted with 0.8 mL 90% methanol and sonicated for 15 min, and the supernatant was transferred into a new tube. The pellet was re-extracted with 0.5 mL of 100% methanol, and the supernatant was combined with the first-step supernatant and dried by vacuum. The pellet was resuspended in 500 µL 0.1 M sodium acetate (pH 5.5) in 10% methanol. An equal volume of 10% TCA was added and the samples

were vortexed and sonicated for 5 min. After centrifugation, the supernatant was extracted three times with 0.5 ml of extraction buffer (ethylacetate/cyclopentane/isopropanol:100/99/1 by volume). After spinning, the organic phases were collected and dried by vacuum. The samples were then dissolved in 250 µL 100% methanol and filtered through a 0.22 µm filter. The samples were then assayed by HPLC-MS/MS analysis on a AB SCIEX QTRAP 4500 system (AB SCIEX, Foster, CA, USA).

## Luciferase complementation imaging assay

*Agrobacterium tumefaciens* GV3101 strain carrying CLuc- or NLuc-tagged consctruct were infiltrated into leaves of 4-week-old *N. b.*. LUC activity in leaves was examined 2 days post infiltration. *N. b.* leaves were treated with 1 µM luciferin and kept in the dark for 5 min to quench the fluorescence. LUC image was captured by CCD imaging apparatus, and the quantitative LUC activity was determined by microplate luminometer. Expression of CLuc-tagged proteins or NLuc-tagged proteins was detected by anti-CLuc or anti-HA western blot, respectively.

## Electrophoretic mobility shift assays (EMSAs)

The full-length CBP60g was cloned into the pGEX-6p-1 vector. VdSCP41C, VdSCP41$_{100-163}$ or VDAG_01962 was cloned into the pET-28a vector. The resulting constructs were transformed into *E. coli* BL21 (DE3) competent cells. BL21 (DE3) strains containing the expression vectors were cultured and induced by isopropylthio-galactoside (IPTG) at 16°C for 16 hr. GST-tagged full-length CBP60g, His-tagged VdSCP41C or VdSCP41$_{100-163}$, and His-tagged VDAG_01962 recombinant protein were affinity purified and used for EMSAs. A 60-bp probe within the DNA fragment 7 (*Zhang et al., 2010*) in the *ICS1* promoter were labeled with [γ-$^{32}$P]ATP using T4 polynucleotide kinase. Binding reactions were carried out in a 20 µl volume of reaction buffer (10 mM Tris-HCl [pH 7.5], 50 mM KCl, 1 mM DTT, 1 µl 50 ng/µl poly[dI-dC]) for 30 min at room temperature. Labeled DNA probe (2 fmol) was incubated with 4 µg; GST-CBP60g. 150 × unlabeled DNA probe was used for competition. 1.5 µg; His-tagged VdSCP41C$_{163-end}$, VdSCP41$_{100-163}$, or VDAG_01962 was preincubated with GST-CBP60g for 30 min at room temperature before DNA binding. The reaction was stopped by adding DNA loading buffer and the samples were separated by a 5% native PAGE gel. After electrophoresis, the gel was autoradiographed.

## RNA isolation and qRT-PCR

Total RNA was isolated with the TRIzol reagent (Invitrogen) and used for cDNA synthesis with Super-Script III First-Strand Synthesis System for RT-PCR (Invitrogen) following the instructions provided by the manufacturer. The quantitative PCR was performed with the SYBR Premix Ex Taq kit (TaKaRa) following standard protocols. *Arabidopsis* Col-0 or transgenic plants were treated with $H_2O$, 1 µM flg22 or nlp20$^{Vd2}$ (*Du et al., 2017*) as indicated for 3 hr. RNA was isolated and used for RT-PCR analysis for the expression of *ICS1* and *FMO1*. *AtTUB4* was used as internal control. To detect gene expression in cotton plants, leaves from the cotton plants were collected 14 days post *Agrobacterium* infiltration. Quantitative RT-PCR was performed as described above. *Gossypium hirsutum HISTONE3* was used as internal control. For RT-PCR analyses of *VdSCP41*, the V592 or Vd∆*scp41* strain was incubated with the roots of 7-day-old *Arabidopsis* Col-0 plants for 2 days. Conidia were collected for RNA isolation and RT-PCR analysis to analyze the expression of *VdSCP41*. For RT-PCR analyses of *V. dahliae*-infected *Arabidopsis* plants, *Arabidopsis* plants infected with or without the V592 or Vd∆*scp41* strain were collected for RNA isolation and RT-PCR analysis for the expression of *ICS1* and *FMO1*. *VdELF1* was used as internal control for *VdSCP41*.

## Fluorescence microscopy

*Agrobacterium tumefaciens* EHA105 strain carrying pCambia1300-35S-CBP60g-GFP, pCambia1300-35S-SARD1-GFP, or pCambia1300-35S-GhCBP60b-GFP was infiltrated alone, or together with the *A. tumefaciens* EHA105 strain carrying pCambia1300-35S-VdSCP41-mCherry (or its variant mutants) into leaves of 4-week-old *N. b.* GFP and mCherry fluorescence were observed with Leica SP8 confocal microscopy 3 days post infiltration. The intensity of fluorescent signals was determined by Image J software. For fluorescence microscopy in *Arabidopsis*, *Arabidopsis* protoplasts were transfected with 35S-CBP60g-GFP alone or together with 35S-VdSCP41-mCherry (or its variant mutants). The protoplasts were incubated overnight under faint light before GFP and mCherry

fluorescence were observed. To examine the subcellular localization of VdSCP41, conidia were cultured on cellophane and incubated for 3–9 days before observation by microscopy. The pieces of cellophane with mycelium were collected and observed as described (*Zhou et al., 2017*). The plasma membrane of the fungi was stained with FM4-64 (red).

## Reporter assay in *Arabidopsis* protoplast

*Arabidopsis* protoplasts were co-transfected with *ICS1::LUC* or *FMO1::LUC* and *35S::RLUC* (Renilla luciferase) alone, or together with *VdSCP41*, CBP60g, or their variants. 12 hr after transfection, the protein of transfected protoplasts was isolated, and the LUC activity was determined by using a Dual-Luciferase Reporter system (Promega) according to the manufacture's instructions.

## Virus-induced gene silencing in cotton plants

The VIGS was performed as described previously (*Gao and Shan, 2013*). Cotton plants were grown at 23–25°C in the growth room until two cotyledons had emerged. A 465-bp fragment of GhCBP60b cDNA was PCR amplified from *G. hirsutum* and cloned into pTRV2 plasmid (*Liu et al., 2002*). The *Agrobacterium* strain carrying pTRV1, together with the *Agrobacterium* strain carrying pTRV2 or pTRV2-GhCBP60b, was infiltrated into the cotyledons of the cotton plants. The cotton plants were root-dip-inoculated with *V. dahliae* V592 2 weeks post *Agrobacterium* infiltration.

## Infection assay

Cotton or *Arabidopsis* plants were infected by the root-dip inoculation method (*Gao et al., 2010*). A conidial suspension of $10^7$/ml from the indicated strain was used as the inoculum. The disease grade was classified as follows: Grade 0 (no symptoms), 1 (0–25% wilted leaves), 2 (25–50%), 3 (50–75%) and 4 (75–100%). The disease index was calculated as 100 × (sum [number of plants × disease grade])/ ([total number of plants] × [maximal disease grade]) (*Xu et al., 2014*). The onion epidermis infection assay was performed as described (*Zhang et al., 2017*). A conidial suspension of $10^7$/ml from the V592-GFP, Vd$\Delta$*scp41*/SCP41-GFP or Vd$\Delta$*scp41*/SCP41$_{-nls}$-GFP strain was inoculated onto the inner layer of onion epidermal cells and incubated on 1% water agar plates for 3–5 days before confocal imaging.

# Acknowledgements

The authors are grateful to Prof. Yuelin Zhang for providing the *Arabidopsis cbp60g-1/sard1-1* mutant, to Prof. Yule Liu for providing the pTRV1 and pTRV2 plasmids, to Prof. Jian Ye for providing the *TPS10::LUC* plasmid, and to Yao Wu for help in SA measurement. This work was supported by grants from the Strategic Priority Research Program of the Chinese Academy of Sciences (XDB11020600) to JZ, the National Natural Science Foundation of China (31730078) to H-SG, the Strategic Priority Research Program of the Chinese Academy of Sciences (XDB11040500) to H-SG, the Youth Innovation Promotion Association of the Chinese Academy of Sciences, the National Natural Science Foundation of China (31571968, 31501593), and the State Key Laboratory of Plant Genomics, Chinese Academy of Sciences.

# Additional information

## Funding

| Funder | Grant reference number | Author |
| --- | --- | --- |
| The Strategic Priority Research Program of the Chinese Academy of Sciences | XDB11020600 | Jie Zhang |
| National Natural Science Foundation of China | 31730078 | Hui-Shan Guo Jie Zhang |
| The Youth Innovation Promotion Association of Chinese Academy of Sciences | | Jie Zhang |

| The Strategic Priority Research Program of the Chinese Academy of Sciences | XDB11040500 | Hui-Shan Guo |
|---|---|---|
| National Natural Science Foundation of China | 31571968 | Jie Zhang |
| National Natural Science Foundation of China | 31501593 | Jun Qin |

The funders had no role in study design, data collection and interpretation, or the decision to submit the work for publication.

### Author contributions
Jun Qin, Kailun Wang, Lifan Sun, Haiying Xing, Sheng Wang, Formal analysis, Investigation; Lin Li, She Chen, Formal analysis, Investigation, Methodology; Hui-Shan Guo, Conceptualization, Supervision, Funding acquisition, Writing—original draft; Jie Zhang, Conceptualization, Supervision, Funding acquisition, Investigation, Writing—original draft

### Author ORCIDs
Jun Qin  http://orcid.org/0000-0003-1306-3433
Jie Zhang  http://orcid.org/0000-0003-2781-8956

### Decision letter and Author response
Decision letter https://doi.org/10.7554/eLife.34902.029
Author response https://doi.org/10.7554/eLife.34902.030

## Additional files
### Supplementary files
• Supplementary file 1. Table S1: data generated from mass spectrum experiments. Data generated from mass spectrum experiment 1 (sheet 1) and mass spectrum experiment 2 (sheet 2). Information of protein IDs, scores, number of peptides for the identified proteins are listed. Matched peptides within CBP60g are shown in red.
DOI: https://doi.org/10.7554/eLife.34902.025

• Supplementary file 2. Primers used in this study. Sequences of primers used in this study.
DOI: https://doi.org/10.7554/eLife.34902.026

• Transparent reporting form
DOI: https://doi.org/10.7554/eLife.34902.027

### Data availability
All data generated or analysed during this study are included in the manuscript and supporting files. Source data files have been provided.

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
