## [Decision Letter]

Thank you for submitting your article "Plant calmodulin-binding proteins are targeted by the *Verticillium* secretory protein VdSCP41 to modulate immunity" for consideration by *eLife*. Your article has been reviewed by three peer reviewers, one of whom is a member of our Board of Reviewing Editors, and the evaluation has been overseen by Detlef Weigel as the Senior Editor. The following individual involved in review of your submission has agreed to reveal his identity: Yuelin Zhang.

The reviewers have discussed the reviews with one another and the Reviewing Editor has drafted this decision to help you prepare a revised submission.

The work by Qin et al. identified V. dahliae VdSCP41 as an intracellular effector that promotes the virulence of the pathogen. VdSCP41 was shown to target Arabidopsis CBP60g and SARD1 and cotton GhCBP60b. It interacts with the C-terminal portion of CBP60g to inhibit its transcriptional activation activity. The Arabidopsis sard1 cbp60g double mutant displayed compromised resistance against V. dahliae and silencing of GhCBP60b compromises cotton resistance to V. dahlia as well. In addition, the contribution of VdSCP41 to the virulence of V. dahliae is significantly reduced in the infection of the sard1 cbp60g double mutant. This work revealed that CBP60g and SARD1 play important roles in resistance to V. dahlia and targeting these two master transcription factors by the effector VdSCP41 has a major contribution to the virulence of V. dahlia. The experiments were well designed, and the findings are novel and very interesting.

A condensed, summary review is provided below. As you can read, in general, the three reviewers recognized the importance of the work and felt that it presented new and interesting information. However, while we recognize that the phenomenon you describe is new, the paper does not really address the actual biochemical function of the VdSCP41 effector, other than its interaction with CBP60g and SARD1. Beyond clearly playing a role in virulence, what is occurring at a mechanistic, biochemical level that explains the action of VdSCP41? Is it a protease, for example? Does binding itself interfere with transcription (e.g., through steric hindrance)? At least some of these possibilities could be examined by simple, straight-forward experiments that, especially considering the magnitude of the work already done, would take relatively little time and effort. Please note that, if the experiments are well designed, even negative answers would be useful. Hence, the reviewers would like to see some additional experiments that could define the actual biochemical function of VdSCP41. The authors should also address the specific comments below, especially those that correct factual errors in the manuscript.

Note specifically the request below to rephrase the title and abstract to reflect the lack of calmodulin binding activity by SARD1.

Summary:

The manuscript presents interesting and novel data clearly implicating a role of the VdSCP41 effector in V. dahliae virulence. However, what is lacking are experiments that better define the biochemical mechanism by which this effector acts.

Essential revisions:

1) Address biochemical mechanism of VdSCP41 action at least to the level of eliminating obvious possibilities. For example, is VdSCP41 a protease that cleaves the transcription factors? Does it bind to the transcription factors and, by so doing, prevent their interaction with their cognate promoters?

2) One of the functions of SARD1 and CBP60g is to promote SA synthesis during pathogen infection. SA levels in transgenic lines expressing VdSCP41 before and after PAMP treatment should be quantified to determine whether VdSCP41 affects PAMP-induced SA biosynthesis. A recent study showed that SARD1 and CBP60g also regulate pathogen-induced pipecolic acid biosynthesis (Sun et al., 2018). The potential contribution of SA and pipecolic acid to resistance against V. dahlia need to be discussed.

3) Biochemical properties of CBP60g and SARD1 are incorrectly described. CBP60g, but not SARD1, binds camodulin. The title and abstract need to be changed accordingly.

4) The manuscript lacks experimental details. For example, how many plants were tested for disease index assays? There is no detail concerning identification of VdSCP41-interacting proteins. How was mass spectrum done? How many repeats? What is the control? It is not clear how many proteins were identified by mass spec. What are the scores and number of peptides identified?

---

## [Author Response]

Essential revisions:

1) Address biochemical mechanism of VdSCP41 action at least to the level of eliminating obvious possibilities. For example, is VdSCP41 a protease that cleaves the transcription factors? Does it bind to the transcription factors and, by so doing, prevent their interaction with their cognate promoters?

We appreciate the reviewers and editors for the insightful advice. We have included new experiments to further investigate the biochemical mechanism between VdSCP41 and CBP60g interaction (Figure 4D and Figure 4—figure supplement 1).

To examine whether VdSCP41 affects the DNA-binding activity of CBP60g, we conducted electrophoretic mobility shift assays (EMSAs) to show that VdSCP41 inhibits the DNA-binding activity of CBP60g (Figure 4D). To examine whether VdSCP41 carries protease activity to cleave or modify CBP60g, we examined the electrophoretic behavior of CBP60g protein both in the absence and presence of VdSCP41 by immunoblot analysis and didn’t find VdSCP41-induced cleavage or mobility change of CBP60g (Figure 4—figure supplement 1). The results thus suggest a binding for inhibition action of VdSCP41 rather than a protease.

2) One of the functions of SARD1 and CBP60g is to promote SA synthesis during pathogen infection. SA levels in transgenic lines expressing VdSCP41 before and after PAMP treatment should be quantified to determine whether VdSCP41 affects PAMP-induced SA biosynthesis. A recent study showed that SARD1 and CBP60g also regulate pathogen-induced pipecolic acid biosynthesis (Sun et al., 2018). The potential contribution of SA and pipecolic acid to resistance against V. dahlia need to be discussed.

We followed the suggestion to measure SA levels in transgenic lines expressing *VdSCP41* before and after *Pst* DC3000 *hrcC^-^* (a nonpathogenic mutant strain which carries a collection of PAMPs) treatment. *VdSCP41*-expressing lines accumulated less free SA in response to *Pst hrcC^-^* treatment than that of non-transgenic WT plants (Figure 2—figure supplement 2A), indicating a suppression of PAMP-induced SA biosynthesis by VdSCP41. We also included sentences (subsection “VdSCP41 Targets CBP60g and SARD1and Interferes with Plant Defense”) to discuss the potential contribution of SA and pipecolic acid to *V. dahliae* resistance as suggested.

3) Biochemical properties of CBP60g and SARD1 are incorrectly described. CBP60g, but not SARD1, binds camodulin. The title and abstract need to be changed accordingly.

We have revised the Title and corrected the statements where are needed. The current Title is “The plant specific transcription factors CBP60g and SARD1 are targeted by a *Verticillium* secretory protein VdSCP41 to modulate immunity.” In the Abstract, the statement has been changed into “The *Arabidopsis* master immune regulators CBP60g and SARD1, and cotton GhCBP60b are targeted by VdSCP41”.

4) The manuscript lacks experimental details. For example, how many plants were tested for disease index assays? There is no detail concerning identification of VdSCP41-interacting proteins. How was mass spectrum done? How many repeats? What is the control? It is not clear how many proteins were identified by mass spec. What are the scores and number of peptides identified?

As requested, we have now included the information of experimental details, including the number of plants for disease index assays and detailed information of mass spectrum analysis, in the corresponding figure legend and Materials and methods section.

Candidate VdSCP41-interacting proteins identified in mass-spec analyses are now listed in Supplementary file 1, in which information of protein IDs, scores, number of peptides and matched peptides are also included. In brief, CBP60g was identified in VdSCP41 immuno-complex in two independent mass spectrum experiments, with multiple matched peptides identified in each experiment. *Arabidopsis* protoplasts transfected with VdSCP45 or empty vector were used as control. CBP60g was not identified in control samples.